# Beliefs and socio-cultural perspectives on hantavirus in a rural community in Panama: An ethnonursing study

Janeth Agrazal Garcia[1]*, Lydia Gordón de Isaacs[2], Elsa Lucia Escalante-Barrios[3], Sergi Fàbregues[4]

1 Faculty of Nursing, Centro Regional Universitario de Azuero, University of Panama, Panama City, Panama, 2 Faculty of Nursing, University of Panama, Panama City, Panama, 3 Department of Education, Universidad del Norte, Barranquilla, Colombia, 4 Department of Psychology and Education, Universitat Oberta de Catalunya, Barcelona, Spain

* janeth.agrazal@up.ac.pa

## Abstract

Hantavirus pulmonary syndrome is a zoonotic disease that has been present in the Americas since 1993 and in Panama since the early 2000s. The disease is transmitted to humans through the inhalation of aerosols containing viruses from feces, urine, and saliva from asymptomatic, infected rodents. While various climatic, ecological, social and cultural factors have been associated with hantavirus transmission, research on beliefs and socio-cultural perspectives regarding its care and prevention remains limited. This study addresses this gap by exploring the beliefs, socio-cultural context and care practices related to hantavirus disease in an endemic community. Guided by the One Health approach and Leininger's Theory of Culture Care Diversity and Universality, an ethnonursing approach was used in a hantavirus-endemic community in Panama from September 3, 2021, to August 25, 2022. The study involved 30 participants, including 11 key participants (community members with personal or familial experience of hantavirus) and 19 general participants (community authorities and health professionals). Data were collected through in-depth interviews, participant observation and field notes. Leininger's four phases of ethnonursing analysis were followed using ATLAS.ti. Five themes were identified: (1) hantavirus develops within the context of the community, with its own expressions, meanings and lifestyles; (2) living in and of the land: what we are and what defines us; (3) hantavirus prevention and its challenges: what we do and what we propose; (4) God, the higher power that helps, protects, and gives strength and resilience in daily life and in the face of hantavirus disease; and (5) sharing within the family: joys, longings, celebrations and concerns about hantavirus. Based on the study findings, we propose a model of culture care decision and action modes for hantavirus prevention, emphasizing the importance of implementing relevant, sustainable, and culturally aligned prevention practices in the affected communities.

**Data availability statement:** Data cannot be shared publicly as participants did not give provide consent for their transcripts to be shared in this manner. The research involved interviews with community members and authorities, as well as with healthcare professionals, discussing potentially sensitive information. Therefore, since the consent statement approved by the Institutional Review Board (IRB) of the Anita Moreno Regional Hospital (Approval No. 38, 27 February 2020), and signed by the participants, did not include the a provision that data would be made publicly available, we do not have participant consent to share thisthese data. However, researchers who meet the criteria for access to confidential data may request access to anonymized data by contacting the IRB of the Anita Moreno Regional Hospital (cbioeticahraam@minsa.gob.pa) or the Principal Investigator Janeth Agrazal (janeth.agrazal@up.ac.pa).

**Funding:** This study was funded by a Competitive Research Fund of the Vice Rector for Research (CUFI-2020) grant awarded to JA by the University of Panama. In addition, JA is a member of the National Research System (SNI) of the National Secretariat of Science and Technology of Panama (SENACYT) and has received research support through this affiliation. The funders had no role in the study design, data collection and analysis, decision to publish, or preparation of the manuscript.

**Competing interests:** The authors declare that no competing interests exist.

## Introduction

Hantavirus pulmonary syndrome is an emerging zoonotic disease found in several countries in the Americas [1]. It poses a significant threat to public health due to its high fatality rate, which can reach up to 50%. In Panama, the first case of hantavirus was reported between 1999 and 2000 in the province of Los Santos, located in the central region of the country, an area that has remained endemic for more than 20 years [2–4]. Hantavirus is transmitted to humans through the inhalation of aerosols contaminated with the faeces, urine and saliva of asymptomatic infected rodents. These infections usually occur in agricultural areas, forests and farms, i.e., places that are the natural habitat of the rodent hosts of the virus. For this reason, people who live or work in these areas are at greater risk of being exposed to the virus and developing the disease [5].

In addition to purely epidemiological factors, the transmission of zoonotic diseases to humans is also the result of multiple complex ecological, climatic, anthropogenic and socio-cultural factors [6]. Human exposure to animals carrying zoonotic diseases is determined by where contact occurs and by the activities and behaviours of people living in that area. These activities are influenced by personal factors (including demographic characteristics such as age, gender, occupation and level of education) and social factors (including cultural elements such as lifestyles, beliefs and traditions, and family and community dynamics) [7]. Therefore, the transmission, prevention and control of zoonotic diseases result from a complex interrelationship between humans, animals and the environment. In some cases, social factors may lead populations to develop beliefs about the causes of these diseases that are inconsistent with scientific evidence and, consequently, individuals may reject or disregard methods recommended by health authorities to prevent and control them [8]. For example, in the Navajo Indian Reservation in the United States, the health system's recommendations for preventing hantavirus were not accepted by the indigenous populations, who did not believe that the disease originated in mice and therefore rejected the hunting of these animals as a control measure because it conflicted with their cultural beliefs [9].

Understanding the nature of the human-animal-environment interrelationship is imperative, as it enables us to adopt an approach to the prevention and control of zoonotic diseases that is both holistic (taking into account the influence of personal and social factors) and collaborative [10–13]. This holistic perspective is consistent with the principles of One Health, a collaborative, multisectoral and transdisciplinary approach that recognises the close links between human health, animal health and the environment [14]. Another approach based on a holistic perspective is Leininger's Theory of Culture Care Diversity and Universality, which argues that health care practices and disease prevention are tied to beliefs, lifestyles, cultural values and norms passed down through generations [15]. This theoretical approach draws from both anthropology and nursing. Anthropology provides an understanding of health and illness from a sociocultural perspective, while nursing uses this understanding to deliver culturally adapted care. The principles of One Health and the Theory of Culture Care Diversity and Universality are essential for developing prevention

programmes that are aligned and consistent with the beliefs held in a particular community, thereby contributing to more effective prevention and control of zoonotic diseases [16].

Several studies have integrated social and cultural perspectives into research on the prevention and control of zoonoses such as malaria [17,18], dengue fever [19,20], leishmaniasis and Chagas disease [21]. In the case of hantavirus, using the Health Belief Model, a social psychological framework, Harris and Armién [22] found that study participants believed, contrary to scientific evidence, that hantavirus transmission was a consequence of fumigation. In another study, Terças et al. [23] found that children from the Haliti-Paresí ethnic group recognised the presence of rodents in the forest as a risk of hantavirus through visual representations in drawings, symbolising the severity of the disease through images of hospitalisation. While these studies have contributed to a better understanding of the socio-cultural dimension of hantavirus, there is still a lack of research that holistically examines the influence of cultural beliefs, lifestyles and social dynamics on hantavirus prevention and control care practices in endemic communities (i.e., regions or specific areas consistently affected by a disease). This gap prevents researchers and professionals from achieving a comprehensive understanding of hantavirus prevention and from designing and implementing effective, context- and culture-specific prevention programmes. This contrasts with the need for nurses and other health professionals involved in hantavirus prevention and control at the community level to be aware of the unique characteristics of each culture and how they influence the behaviours people adopt to prevent the disease.

A thorough understanding of socio-cultural factors and their interaction with people's perceptions of hantavirus is essential for the development of prevention programmes that are comprehensive, holistic and socially accepted [24]. This understanding should incorporate insights from the affected community regarding their care practices, the beliefs associated with these practices, and the socio-cultural context of the community. Such an understanding would increase the effectiveness of prevention programmes by tailoring them to specific contexts and cultures, and would also help to address beliefs that are inconsistent with scientific evidence about the causes of the disease and how to prevent it. Thus, building on Leininger's Theory of Culture Care Diversity and Universality, this qualitative ethnonursing study aimed to explore, from a holistic perspective, the relationships between beliefs, social and cultural context, and care practices related to hantavirus disease among people living in an endemic community.

## Materials and methods

### Design

This study was part of a larger explanatory sequential mixed methods study, which consisted of an initial quantitative phase based on a survey, followed by a qualitative phase aimed at providing deeper insight into and explanation of some of the findings from the initial quantitative phase [25]. This article reports on the qualitative phase, which was conducted from September 3, 2021, to August 25, 2022, in a community located in the district of El Bebedero, within the province of Los Santos, Panama. This phase employed an ethnonursing approach, a method developed by nurse and anthropologist Madeleine Leininger that seeks to explore beliefs, experiences, and cultural practices associated with health and illness through a naturalistic, open, and inductive approach. The results of the quantitative and mixed methods phases have been published elsewhere [26,27]. The research team included researchers with expertise in ethnonursing (JAG [MSN, RN]; LDI [MSN, PhD, RN]), public health (JAG), and qualitative and mixed methods research (JAG, EE [PhD], SF [PhD]).

### Immersion in the field of study

The study was conducted in a rural agricultural area considered endemic for hantavirus, located in the province of Los Santos in central Panama. Prior to qualitative data collection, the principal investigator (JAG), who was a PhD student at the time the study was conducted, immersed herself in the field with the help of a community leader. This was done in order to gain acceptance from members of the cultural group who had participated in the previous quantitative phase, earn their trust, and obtain more accurate and reliable data [28–30]. During this process, she visited households,

community institutions, and health facilities. In addition, she adopted a learner's mindset, acknowledging members of the cultural group as the primary sources of knowledge (15), and developed skills such as maintaining an openness to learning from others, and perceiving and listening with respect and a willingness to understand [31].

The principal investigator, who is familiar with the epidemiological data on hantavirus morbidity and mortality and its status as a public health problem in the study area, completed the immersion process with the support of a female community leader over a period of three months. Leininger's Stranger to Trusted Friend Enabler tool was used to determine the appropriate time to begin data collection. This tool contains a set of qualitative, observable indicators that allow researchers to assess participants' trust in them. This trust is expressed through observable behaviours, emotional expressions, and a willingness to communicate. Once trust is established, researchers transition from being strangers to trusted friends, thereby facilitating the collection of more accurate and meaningful data [29]. The transition from stranger to trusted friend for the principal investigator was perceived differently within the cultural group. While some participants demonstrated trust in the researcher within the first few weeks — through behaviours and gestures such as invitations to return and gifts of agricultural produce and plants— others were more reserved and cautious. Based on these observations, interviews were initiated with participants who had shown greater trust.

## Participants

In line with the ethnonursing method, two types of participants were selected: key participants (n = 11) and general participants (n = 19). Key participants included local community members with personal or familial experience with hantavirus. General participants included health professionals skilled in preventing and treating hantavirus, as well as community authorities knowledgeable about the virus and the social dynamics of the area. Including both types of participants was essential to achieving a more comprehensive understanding of the disease, as it allowed us to consider not only the perspectives of those directly affected but also the insights of health professionals and community authorities.

Eligible key participants were members of the community who had participated in the first, quantitative phase of the study and completed the survey. To ensure diversity among key participants, individuals to be contacted were selected based on age, gender, education, personal or family history of hantavirus, and length of residence in the community (at least one year). Diversity of key participants was achieved in all criteria except gender, as most men who were contacted refused to participate due to time constraints. Eligible general participants were health professionals and community authorities with at least one year's experience working in the study area, selected on the basis of age, gender and professional role within the community. Diversity of general participants was achieved in gender and age. In terms of their role, most were health professionals.

Potential key and general participants were contacted by the principal investigator and the community leader either by telephone or by visiting their homes or workplaces. They were informed of the purpose of the study, given details of what their involvement would entail, and asked if they were willing to participate. The number of key participants was determined by the criterion of theoretical saturation, i.e., when the recruitment of new participants did not yield any additional ideas or concepts beyond those already identified with previous participants [32]. The number of general participants was based on the ethnonursing method, which recommends approximately two general participants for every key participant in order to achieve an in-depth understanding of the phenomenon under study [15]. There were no differences in the distribution of key participants according to the sampled variables, except for gender: only one of the 11 key participants was male, as the other men contacted declined to participate due to lack of time. The distribution of general participants was homogeneous with respect to the sampled variables. Key and general participants who agreed to participate in the study were visited at home or at work. During this visit, the researchers provided further information about the study and the participants signed an informed consent form.

## Data collection and analysis

The four phases of data collection and analysis proposed by Leininger for ethnonursing [15,33] (illustrated in Fig 1) were carried out iteratively, moving back and forth between them, guided by the research question. The first phase involved the collection and transcription of data. Data were collected through in-depth interviews, observations and field notes. Interviews, which were audio recorded, lasted between 30 and 60 minutes for key participants and between 15 and 30 minutes for general participants. The in-depth interview questions were developed based on the Sunrise Enabler, a cognitive map from Leininger's Theory of Culture Care Diversity and Universality [33] (see S1 File Interview Guide). Using this map allowed the interview questions to explore cultural factors, worldviews, and social and environmental contexts related to hantavirus, as well as strategies for its care and prevention.

The principal investigator conducted interviews with key participants in their homes, and with general participants in their offices or at the community health facility. The interviews were conducted with only the participant and the researcher present. Interviews were transcribed using the NVivo Transcription tool. To maintain confidentiality and anonymity, no personally identifying questions were asked and each participant was given an anonymous identification code consisting of a letter identifying the type of participant (K for key participant and G for general participant) and a sequential number (K01/G01). After the interviews were transcribed and checked for accuracy, the audio recordings were deleted. Observations were conducted according to Leininger's Observation-Participation-Reflection Enabler [29], which guides the researcher through a progression from observation and listening to a more active role involving participation and reflection [33]. Observations were made in family, environmental and social settings related to hantavirus care and prevention and documented in field notes taken using the iPad Notes application. These notes recorded aspects of the environmental and social settings, expressions, interactions with participants, and reflections by the researchers. The notes provided contextual depth to the interview data, thereby enhancing and strengthening the interview analysis process. The first phase concluded on February 18, 2022.

In the second phase, descriptors (direct expressions of the participants) were identified from the data collected in the first phase, and inductive codes (meaningful phrases or expressions assigned to the descriptors by the researcher) were created. These codes were then analysed for similarities and differences between the participants. In the third phase, the previously identified codes were examined to identify recurring patterns of expressions, meanings, structural forms or interpretations. The fourth phase, characterised by a higher level of abstraction and critical analysis, consisted of synthesising the patterns identified in the third phase and formulating themes based on them. Each formulated theme was also reviewed to ensure that it was grounded in the data or findings, and recommendations were made for developing culturally congruent prevention strategies (including a model of culture care decision and action modes for hantavirus prevention) [15,30,33]. These recommendations are detailed in the implications of the study for nursing. The procedures followed in phases two through four were initially carried out by the principal investigator and subsequently reviewed by the other members of the research team (LDI, EE, and SF). Analysis was performed using ATLAS.ti v.23 software.

## Quality

Throughout the study, several strategies were implemented to meet the quality criteria proposed by Leininger [34]: credibility, confirmability, meaning-in-context, saturation and transferability. First, to ensure credibility, a three-month immersion in the field was undertaken and the Stranger to Trusted Friend Enabler was used to assess the level of trust gained from the study group. In addition, the results of the key and general participant interviews, observations and field notes were triangulated, which not only strengthened the findings but also provided a more thorough understanding of the phenomenon under study. The findings were also discussed among the research team to assess their credibility and consistency. Second, the criterion of confirmability was addressed by repeating key and general participants' statements at different points during the interviews. For example, if a participant introduced a particular theme, the researcher would return to it later in

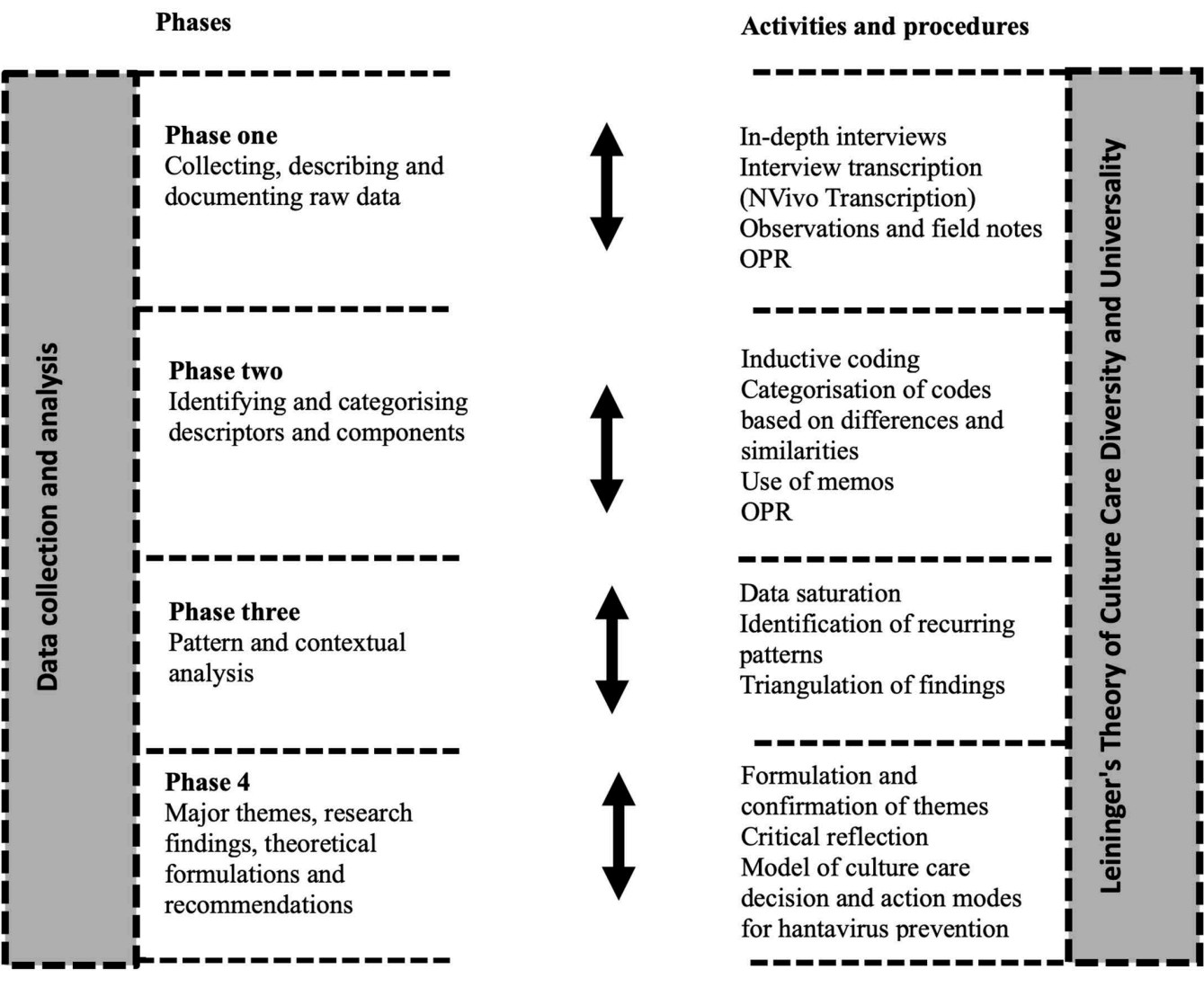

**Fig 1. Phases of Ethnonursing Data Collection and Analysis.**

the same interview or in a follow-up interview to corroborate the data. Third, to meet the criterion of meaning-in-context, and thereby ensure that the data had a special meaning within the participants' lived experience, the principal investigator paid particular attention during the interviews to the importance that key and general participants attached to certain aspects of life. For example, emphasis was placed on the value of agriculture as a way of life and livelihood passed down through generations of families, and the role of family and community support in daily life and in the context of hantavirus disease. Fourth, data saturation was reached when no new patterns or themes were identified in the analysis of the interview data. In line with the iterative nature of the design used, interviews and data analysis were conducted concurrently, with each interview being analysed immediately after it was conducted. As the analysis progressed, the findings informed the selection of new participants who could provide further information on particular patterns, while still adhering to the previously described participant selection criteria. When information was repeatedly expressed by multiple participants and no new patterns were identified in the data analysis, recruitment of new participants was stopped. Finally, to support the transferability of the results to other settings, the key and general participants, study context and methods were described in detail. This level of detail should enable other researchers to assess the applicability of the study's findings and conclusions to similar contexts. The reporting of the study methods and results followed the Consolidated Criteria for Reporting Qualitative Research (COREQ) guidelines [35] (see S2 File COREQ Checklist).

### Ethical considerations

The study protocol was reviewed and approved by the Institutional Review Board of the Anita Moreno Regional Hospital (part of the University of Panama), located in the province of Los Santos, Panama (Approval No. 38, 27 February 2020). All participants were informed about the purpose of the study. They were also informed that the interviews would be recorded, that their data would be kept confidential, and that they could withdraw from the study at any time. After receiving this information, those who agreed to participate signed an informed consent form.

### Inclusivity in global research

Additional information regarding the ethical, cultural, and scientific considerations specific to inclusivity in global research is included in S3 File Inclusivity in Global Research Questionnaire.

## Results

### Participant characteristics

The characteristics of the key and general participants are shown in Table 1 (see S4 File Key and General Participants Characteristics). Of the 11 key participants, 10 were female and ranged in age from 33 to 83 years. Eight reported having a close family member with a history of hantavirus, and three reported a personal history of the disease. Two key participants had no formal education, two had completed primary school, three had completed secondary school and four had a university degree. Seven key participants reported that they had lived in the community all their lives or for more than 20 years, while four reported that they had lived in the community for between 10 and 15 years. Of the 19 general participants, 11 were female. General participants ranged in age from 29 to 66 years. Fourteen were health professionals in the public service, while the others were community authorities from public and educational institutions.

### Themes and patterns

Table 2 lists the themes and patterns that were identified in the interviews. The following five themes were identified: (1) hantavirus develops within the context of the community, with its own expressions, meanings and lifestyles; (2) living in and of the land: what we are and what defines us; (3) hantavirus prevention and its challenges: what we do and what we

**Table 1. Key and general participant characteristics.**

| | Key participants (n = 11) | General participants (n = 19) |
|---|---|---|
| Gender, n (%) | | |
| Male | 1 (9.0%) | 8 (42.1%) |
| Female | 10 (90.9%) | 11 (57.8%) |
| Age range years, n (%) | | |
| ≤ 30 | 0 (0%) | 1 (5.3%) |
| 31-40 | 1(9.0%) | 9 (47.3%) |
| 41-50 | 4 (36.4%) | 5 (26.3%) |
| 51-60 | 4 (36.4%) | 1 (5.3%) |
| >60 | 2 (18.2%) | 3 (15.8%) |
| Education, n (%) | | |
| Non education | 2 (18.2%) | N/A |
| Primary | 2 (18.2%) | N/A |
| Secondary | 3 (27.2%) | N/A |
| University | 4 (36.4%) | N/A |
| Personal or family history of hantavirus | | |
| Family | 3 (27.2%) | N/A |
| Personal | 8 (72.8%) | N/A |
| Length of residence | | |
| Entire life | 5 (45.4%) | N/A |
| 20-25 years | 2 (18.2%) | N/A |
| 15-19 years | 2 (18.2%) | N/A |
| 10-14 years | 2 (18.2%) | N/A |
| Role | | |
| Health professional | N/A | 14 (73.7%) |
| Community authority | N/A | 5 (26.3%)) |

Note. N/A = Not Available or Not Applicable.

propose; (4) God, the higher power that helps, protects, and gives strength and resilience in daily life and in the face of hantavirus disease; and (5) sharing within the family: joys, longings, celebrations and concerns about hantavirus.

**Theme 1: Hantavirus develops within the context of the community, with its own expressions, meanings and lifestyles.** This theme comprises four patterns that describe the "emic" perspective on hantavirus in a community that has been endemic for over two decades. Participants described the experience and the social and cultural significance of hantavirus illness beyond the disease itself and its symptoms and complications.

*Pattern 1.1: Different beliefs about the mode of hantavirus transmission.* Key participants described three beliefs about possible modes of hantavirus transmission. The first belief is consistent with the scientifically established and empirically supported mode of transmission, namely via a vector: *"He got infected there because there were mice in the classrooms... there were mouse droppings and urine"* (K07). The second belief links the disease directly to the fumigation of agricultural crops, a view not supported by scientific evidence: *"Her evil* [referring to hantavirus] *was the fumigation. Yes, because the fumigation affected a lot of people"* (K01). The third belief, also inconsistent with scientific evidence, is based on the idea that the rodent (i.e., the disease vector) is contaminated by the chemicals used in fumigation, causing hantavirus: *"...now they're using too many chemicals, too many chemicals... When the mouse eats the rice and the maize, of course its urine is going to be infected by this chemical... So that's now bringing us this mouse with that disease"* (K04).

**Table 2. Themes and patterns identified in the data analysis.**

| Themes | Patterns |
|---|---|
| 1. Hantavirus develops within the context of the community, with its own expressions, meanings and lifestyles | 1.1 Different beliefs about the mode of hantavirus transmission |
| | 1.2 Being ill with hantavirus is hard and sad |
| | 1.3 Hantavirus is here and affects us all |
| | 1.4 Sometimes people do not want to seek care, and we are not prepared |
| 2. Living in and of the land: what we are and what defines us | 2.1 Agriculture and rural life: our way of living, working and surviving |
| | 2.2 We are hardworking, cheerful and good-natured people |
| | 2.3 We struggle together and support each other in daily life and in the face of hantavirus |
| 3. Hantavirus prevention and its challenges: what we do and what we propose | 3.1 Belief that it is difficult to prevent hantavirus and eliminate mice |
| | 3.2 What we do to prevent hantavirus |
| | 3.3 Hantavirus risk factors and suggested preventive measures |
| 4. God, the higher power that helps, protects, and gives strength and resilience in daily life and in the face of hantavirus disease | 4.1 Belief in the power of God, trust in his help and protection in daily life and in the face of hantavirus |
| | 4.2 Belief in prayer as a form of communication with God, who offers help, strength and resilience in the face of hantavirus |
| 5. Sharing within the family: joys, longings, celebrations and concerns about hantavirus | 5.1 Sharing and living together as a family: remembering those who are no longer here or who are far away |
| | 5.2 The value of family care in the context of hantavirus |

*Pattern 1.2: Being ill with hantavirus is hard and sad.* Key participants described the signs and symptoms of hantavirus disease, emphasising that there are both mild and severe forms of the disease. They expressed fear and anguish about the disease: *"I'm telling you, I'm really scared of hantavirus... It's like something big. Really, I don't know"* (K09). These participants spoke of weakness, fatigue, weight loss and fever as aftereffects of the disease that prevented them from immediately returning to normal life: *"And when I came out of hospital... it took me six months, six months to recover very slowly. I was down to 100 pounds, just terrible, recovery is harder"* (K08). General participants, mainly health workers, described the symptoms experienced by people with hantavirus when seeking medical care: *"When she got here, she was so weak all over she couldn't hold herself up"* (G05).

*Pattern 1.3: Hantavirus is here and affects us all.* Key participants noted that hantavirus affects their daily lives; for example, the suspension of social activities in the community due to rising case numbers, which also has an economic impact: *"When they cancelled the carnival. Oh my God, that was huge because never, honestly, never, never had they cancelled an event of that size"* (K06); *"Or when a fair suddenly comes along, they stop making money from it too—yes, financial income"* (K02). General participants, especially health workers, acknowledged that community members are inconvenienced and bothered by the hantavirus-related health interventions: *"They were protesting because, thanks to the famous hantavirus mouse, their income was going to drop when the carnivals and the fair were cancelled"* (G02).

*Pattern 1.4: Sometimes people do not want to seek care, and we are not prepared.* Key participants described how they delay seeking care due to apprehension about the illness or because they confuse the early symptoms of hantavirus, such as fever, with those of other illnesses: *"Well, when she had a fever, she didn't want to go to hospital"* (K09). In

many cases, it is the family who intervenes and advises the person to see a doctor: *"And then she said she hadn't gone to hospital, that there were a lot of people there... And I said, 'oh just go, go'"* (K03). They also reported that despite the hantavirus situation in the community, the hospital is not well-equipped to deal with severe cases: *"Here the hospital is not prepared; it's where there've been the most hantavirus cases in the district and the hospital is not prepared"* (K04).

**Theme 2: Living in and of the land: what we are and what defines us.** This theme comprises three patterns that broadly describe the sense of belonging to the community, the symbols that represent it, and the interpersonal and social relationships formed within it. These community relationships convey a shared past, present and future shaped by the hantavirus phenomenon.

*Pattern 2.1: Agriculture and rural life: our way of living, working and surviving.* Key participants described agricultural and rural work as an important part of the family and community economy for generations: *"As long as I can remember I've always seen it that way; people have always worked in agriculture"* (K07); *"My father always worked in agriculture and livestock; he used to milk animals and do that sort of thing. After my father died, we, we continued with that"* (K11); *"I used to work in agriculture, then I came here and kept doing the same thing, agriculture… because I have always liked working in agriculture every day"* (K01). Some participants described this work as a family commitment and a source of pride and affection: *"My husband planted maize and we're all in it; it's a family thing"* (K06); *"...because I've liked farming every day"* (K01). General participants described agriculture as the main source of employment in the community: *"Well, Bebedero is a place of work and rural life. People here are farmers"* (G16). Field observations identified large rice and maize fields, as well as workers, farming equipment, tools and materials used in agriculture.

*Pattern 2.2: We are hardworking, cheerful and good-natured people.* Key participants described themselves as hard-working, cheerful people who enjoy parties and music: *"They do a lot, working out there, always. They also really like to party"* (K03); *"They love a party, dear God, so much, too much"* (K02). Religious celebrations, as well as celebrations involving partying, drinking and dancing, were mentioned by key participants in general terms, with no particular emphasis on one type of event over another: *"...celebrate, at least the village does, [festivals] like Carnival, the Day of the Farmer, Saint Roch—who is the co-patron saint—and Candlemas"* (K04). General participants identified this love of celebration as a hallmark of the community: *"...it's a very festive community"* (G02). Both the strong work ethic and the community's enthusiasm for fun and celebration were confirmed by field observations. For example, early in the morning people could be seen starting their work in the crop fields, and between tasks they would sing or cheerfully greet others arriving at the fields. Various recreational areas with music were also observed, located along the riverbanks or in the centre of the community.

*Pattern 2.3: We struggle together and support each other in daily life and in the face of hantavirus.* Key participants described the struggle to ensure the well-being of the community and the solidarity among its members, especially among neighbours: *"You can see it in neighbours looking out for each other"* (K05). The mutual care among community members was also evident in the concern shown for those who had contracted hantavirus: *"And people from over there, from where I live... and from here in Bebedero, people came to see me when I was sick"* (K08). General participants also acknowledged the community's spirit of solidarity: *"People tend to be very sociable. They look after their neighbours. They go and see how those with hantavirus are doing"* (G19); *"There is a lot of human solidarity in the community"* (G17). During field observations, it was noted that sick people and older adults living alone received visits and care from other members of the community.

**Theme 3. Hantavirus prevention and its challenges: what we do and what we propose.** This theme comprises three patterns that describe the meanings attached to hantavirus prevention and control within the community, as well as social and cultural perspectives on risk and vulnerability. These processes of prevention, control and risk are described by participants in terms of their feasibility and applicability in everyday life.

*Pattern 3.1: Belief that it is difficult to prevent hantavirus and eliminate mice.* Key participants—both those whose beliefs about the causes and modes of transmission agree with scientific evidence and those whose beliefs do

not—emphasised the challenges of preventing hantavirus in agricultural working environments. The main challenge identified by both groups was the proximity of rice and maize crops to homes, which increases the presence of rodents in domestic spaces: *"A lot of maize, rice and sugar cane are planted around here, and they're planted near the house"* (K03); *"And how are we supposed to stop this animal from coming into the house? We have no way"* (K04). However, those who believe that hantavirus is directly linked to fumigation (a cause not supported by scientific evidence) expressed doubt that eliminating mice would reduce the prevalence of the disease: *"You can get rid of it—I mean the mouse—but the hantavirus will still be there"* (K02). This view is closely associated with the belief of these participants that mice have been present in fieldwork for many years, during which they were not regarded as a risk factor. Another point raised by key participants, and one that adds to the difficulty mentioned above, is the limited cooperation of some community members in preventing the disease: *"Well, we try to clean our gardens and such, but sometimes the neighbours don't all clean theirs"* (K11). General participants noted that it is difficult to prevent hantavirus if people do not believe in the scientifically established mode of transmission: *"If someone doesn't firmly believe that the mouse is the cause, they're very unlikely to make any changes"* (G03).

*Pattern 3.2: What we do to prevent hantavirus.* While a full commitment to actively prevent hantavirus among key participants was lacking, several preventive measures were mentioned. Among these, maintaining cleanliness inside and outside the house was most often cited, with more emphasis on the inside: *"We look after ourselves through cleanliness, not throwing food in the garden, trying to keep things clean inside the house, everything clean"* (K04). This measure was predominantly mentioned by key participants who believe that hantavirus is transmitted by the rodent vector and by those who believe that transmission occurs through contamination of mice by chemicals used in fumigation (see Pattern 1.1). Participants from both groups also described other preventive measures, such as using traps and poison to kill the mice, keeping a pet cat to hunt them, and storing harvested crops in lidded containers to prevent them from entering, although the latter came up only rarely. None of these measures were mentioned by key participants who believe that hantavirus transmission is associated with the process of fumigating agricultural crops. Some of these participants also insisted that eliminating the vector would have no impact on disease prevention: *"You can get rid of it—I mean the mouse—but the hantavirus will still be there"* (K02). General participants confirmed that the most common preventive practice is to maintain household cleanliness: *"In terms of practices, people regularly clean their houses, the inside of their houses"* (G12). Both key and general participants described the educational programmes on hantavirus prevention conducted by health workers. The main aim of these programmes was to raise community awareness of the importance of keeping the home and its surroundings clean and avoiding contact with rodents as key preventive measures: *"They've talked to us about hantavirus. They've come here, spoken, given us a leaflet, but they've not asked us, or rather told us, how to keep the house"* (K04); *"...going door to door, house to house, talking to people about how to clean and everything"* (G02). Field observations confirmed that most key participants kept the outside of their homes clean, while the practice of storing harvested crops in lidded containers was rarely observed.

*Pattern 3.3: Hantavirus risk factors and suggested preventive measures.* Key participants rarely described the risks of hantavirus transmission, whereas general participants referred to them frequently. Key participants primarily associated risk with activities related to rural and agricultural work. For example, some of them identified the proximity of crop fields to their homes as a risk because of the presence of the rodent vector in these areas. This proximity facilitates the vector's entry into homes and increases the likelihood of inhaling aerosols contaminated with rodent urine, droppings or saliva—currently the most scientifically supported mode of hantavirus transmission. Another farming-related risk mentioned by other participants was fumigation, but this is not supported by scientific evidence. General participants highlighted risks of hantavirus transmission due to the infrequent use of protective measures (e.g., gloves, masks) in situations where aerosols contaminated with rodent urine, faeces and saliva may be inhaled. Such situations include handling live or dead rodents and entering enclosed spaces such as warehouses, sheds or unoccupied houses where the vector may be present: *"A young man came in—he had gone into the grain store without any protection—and I said to him, 'you mean you*

*didn't even wear a mask?' He told me no, that generally people don't do that here"* (G09). Other hantavirus risk factors reported by general participants included the storage of harvested crops inside the home, the proximity of cultivated fields to dwellings, and the placement of food for domestic animals (chickens) in the garden. These practices encourage the rodent vector to enter homes in search of food and to leave behind urine, saliva or faeces, thereby increasing the risk of hantavirus transmission.

In addition to explaining the measures they currently take to prevent hantavirus (Pattern 3.2), both key and general participants also mentioned measures that they thought should be taken to prevent hantavirus in the community context. Key participants suggested actions that were consistent with their specific belief about the cause of hantavirus, among the three described earlier (in Pattern 1.1). As highlighted above, the first of these beliefs is consistent with scientific evidence, whereas the other two are not. Key participants who hold the first belief (that the rodent is the disease vector) expressed the need to avoid planting crops near homes to reduce the presence of rodents: *"...they shouldn't plant near the houses, that's what I think... because if they plant, say, maize or sugar cane… And they say they* [the mice] *really like sugar cane too"* (K03). Those who hold the second belief (that fumigation is directly responsible for the disease) suggested reducing the use of fumigation in agriculture: *"The handling of agrochemicals, avoiding the handling of chemicals... That fumigation business doesn't do anything anyway"* (K02). Finally, key participants with the third belief (that the rodent inhales fumigation chemicals and then transmits the disease to humans) suggested a combination of measures, some in line with scientific evidence, such as eliminating the rodent, and some not, such as controlling fumigation: *"The mouse has to be eliminated, just in case, because you never know what… Honestly, I don't know if it's this or that, but for me it's that fumigation poison"* (K10). General participants, including health professionals and authorities, suggested science-based measures such as proper storage of harvested crops and the use of masks or protective equipment when entering enclosed spaces or disposing of dead rodents. They also stressed the importance of working with community leaders and schools to strengthen community outreach.

**Theme 4: God, the higher power that helps, protects, and gives strength and resilience in daily life and in the face of hantavirus disease.** This theme comprises two patterns that describe the importance of spirituality and religion in the context of the disease and in decision-making during this process. Spiritual or religious beliefs are expressed through symbols and meanings that make sense of the disease, its risks and its consequences.

*Pattern 4.1: Belief in the power of God, trust in his help and protection in daily life and in the face of hantavirus.* For key participants, the well-being of the community and each of its members depends on God. For example, when asked how they were doing, they replied: *"Fine, thank God"*. God's help is present in everyday life and contributes to positive outcomes. As the participants described: *"...with God's help everything went well"* (K08); *"...we'll harvest next month, God willing"* (K01). This divine help was also described as a means of coping with hantavirus disease. Key participants talked about how God's protection helped them or their relatives against the disease and its complications: *"And well, it was thanks to God that I didn't end up in intensive care"* (K02); *"But thanks to God she got better"* (K06). Meanwhile, general participants acknowledged that various religious activities and celebrations take place in the community: *"People enjoy Candlemas... Saint Roch is commemorated... and so is Saint Isidore the Labourer"* (G17).

*Pattern 4.2: Belief in prayer as a form of communication with God, who offers help, strength and resilience in the face of hantavirus.* Key participants described prayer, whether individual or communal, as a resource against hantavirus disease: *"And I pray to Jehovah there and say I don't want my daughter to get hantavirus"* (K05); *"They also went to church to pray for me"* (K08). These prayers were described as a source of comfort and strength: *"...and God gave me strength"* (K09); *"I asked the Lord to give me strength and thank God we got through it"* (K03). During the interviews, when key participants expressed their trust in God, various religious symbolic gestures were observed, such as looking up and raising their hands to the sky or praying aloud. Religious images and representations were also seen displayed in their homes.

**Theme 5. Sharing within the family: joys, longings, celebrations and concerns about hantavirus.** This theme comprises two patterns that describe the role of the family as a social unit and its response to health risks. Each family has unique characteristics that define how it functions and how members relate to one another. Within the family, lifestyles, values and practices that can either support or jeopardise health are passed on from one generation to the next.

*Pattern 5.1: Sharing and living together as a family: remembering those who are no longer here or who are far away.* Key participants expressed the view that family is not limited to blood relatives (parents, siblings, children, aunts and uncles, etc.), but also includes the members of their partners' families: *"Whenever something happens, the family on my husband's side and on my side always call me and say, 'Hey, we need this' or 'How do we do this? How do we do that?'"* (K06); *"Oh my God, my family is huge. In this house there's just a few of us... but on my dad's side it's a big family and on my mum's side too..."* (K11). The family was described as the nucleus for everyday gatherings such as walks to the river or the beach, family get-togethers, local celebrations and reunions at funerals when a relative dies: *"During Carnival, Holy Week... we all get together as a family"* (K10); *"We see each other for birthdays, but we all get together for wakes"* (K07). Key participants also talked about remembering relatives who live far away or who have died: *"I have two granddaughters, one is one and a half years old, and the other is two... I don't know what to do without either of them around... Now that they're gone, I feel more alone"* (K09). General participants acknowledged the existence of strong bonds of companionship and support between family members: *"Families are close and have strong ties"* (G17).

*Pattern 5.2: The value of family care in the context of hantavirus.* Key participants explained how families provide unconditional care to their members when there is fear of a possible hantavirus diagnosis, as well as when the disease has been confirmed. Families guide and encourage their members to seek medical care when hantavirus is suspected: *"My family kept telling me, 'Go, don't wait, you're asthmatic and it's going to be hantavirus'"* (K02). Once the person has been diagnosed, the family takes on various caring roles, such as visiting the patient in the hospital and staying with them for extended periods, shouldering household responsibilities for the sick person or their hospital companion and providing spiritual support through individual and communal prayers: *"We never left her side. I was in the intensive care ward for 13 days. My mother was in a hammock, my youngest daughter in a chair, and me... I felt that if I left, she wouldn't be there when I came back"* (K09). All caregiving described was associated with the time of diagnosis and the course of disease; in contrast, very few examples of care aimed at prevention were cited. General participants also recognised the importance and value of family support during illness: *"I felt miserable, just absolutely miserable. My body ached terribly, and I had a fever... So I went to my brother's and he helped me"* (G15).

## Discussion

### Summary of findings and relationship to previous literature

This ethnonursing study aimed to explore the beliefs, social and cultural context, and care practices related to hantavirus disease among people living in an endemic community in Panama. Five themes were identified, covering aspects such as beliefs about the disease, the social and financial impact of hantavirus, preventive practices, and the importance of context, family, and social and spiritual support in relation to both the disease and its prevention.

Three beliefs about hantavirus transmission were identified among the participants. The first is consistent with the mode of transmission described in the scientific literature, i.e., via a rodent vector, while the other two are not, being directly or indirectly related to agricultural fumigation processes. These last two beliefs align with the findings from a previous study conducted in this endemic community [22]. These two beliefs, not consistent with scientific evidence, may hinder the adoption of effective measures to control the vector [36]. Such measures are currently the primary preventive strategy against hantavirus in the absence of other strategies, such as vaccination [37] and planting crops at a distance from homes. When community beliefs and knowledge about hantavirus diverge from scientific evidence, barriers to implementing effective preventive and control measures can arise. This divergence can lead to a lack of trust in health

authorities, low adherence to or rejection of official recommendations, and a preference for traditional practices and forms of care, such as the use of medicinal plants and other symbolic interpretations of the disease, as reported in previous studies on hantavirus [9] and other zoonoses [17,18]. Understanding how communities interpret and address hantavirus is key for designing preventive strategies that are more culturally sensitive, participative, and aligned with local realities.

In this study, community members recognised hantavirus as a serious disease and were able to identify its signs, symptoms and complications. They also expressed fear, uncertainty and distress about the personal risk of becoming ill, dying or losing family members and friends to the disease. This finding is consistent with the qualitative study by Terças et al. [23], in which children from the Haleti-Paresí ethnic group in Brazil were aware of the rodent disease vector and the environments in which it was found (inside and outside the forest). In addition, these children conveyed their perception of the severity of hantavirus through drawings depicting hospitalisation. Similarly, the endemic persistence of hantavirus in the study area for over 20 years [2–4], along with the associated risk of death [4], has led to a constant perception of risk and threat among the population. This climate of risk may be exacerbated by personal and familial experiences of hantavirus, as described in the narratives of our study participants, and passed down through generations.

From an anthropological perspective, disease involves both objective and subjective expressions. The objective expression of disease is related to biological aspects such as signs and symptoms, while the subjective expression considers the meaning and lived experience of disease for the individual and the social group. Disease has a particular meaning for individuals, determining the extent to which they seek to prevent, cure or treat it. Recognising disease as something undesirable that causes discomfort, suffering, anguish or death is essential for both individual and community prevention [38].

The results of this study show that the participants recognised the impact of the disease on the lives of individuals and their families, as well as its serious consequences. Participants also described how, when hantavirus cases rise, health measures or restrictions are implemented, which, first, negatively affect income and the realisation of social activities and, secondly, highlight the limitations of the health system in disease prevention. This is in line with Menéndez [39], who argues from an anthropological perspective that diseases are not only biological processes, but also cause social, economic and political problems. In this study, we found evidence that hantavirus disrupts these three dynamics. Socially, it impacts deeply ingrained social celebrations. Economically, it limits the generation of resources due to the suspension of economic activities. Politically, it highlights inequalities in access to healthcare and increases perceptions of social injustice.

The study findings reveal two important aspects within the community affected by hantavirus: the symbolic significance of community celebrations and the profound cultural roots of agriculture. Celebrations were portrayed as significant venues for socialising, cultural expression, and strengthening community links. Carnival festivities and fairs are, for example, important social events that bring families, friends, and the community together. Likewise, agriculture was not merely described as a source of sustenance and income, but also as an important tradition passed down through generations. Participants explained that agricultural skills are taught within the family unit, involving all its members. The familial nature of agriculture, together with the collective pride associated with it, demonstrates how important it is in reinforcing a sense of belonging and enhancing social cohesion.

These findings are in line with Tobasura Acuña and Obando Moncayo [40], who noted that for rural workers, farmers and indigenous groups, land is valued not only for its utility in producing food and generating income and power, but also for its emotional and even sacred significance. In this regard, anthropologist Heckadon Moreno [41], referring to the province of Los Santos and the district of Tonosí, described agriculture as a family enterprise that provides food for its members and serves as a source of income. Health measures aimed at preventing hantavirus, such as adjusting agricultural practices to increase the distance between crops and dwellings, may affect the cultural role of agriculture in the everyday life of the community members. Additionally, celebrations provide important opportunities for community gatherings, cultural expression, and the strengthening of community bonds. However, these events may also be affected by the hantavirus prevention health measures. Therefore, respectful dialogue between community members and health professionals is essential to effectively implement and accept these measures.

A majority of key participants reported that the most common way they prevent hantavirus is by maintaining cleanliness in and around the home. They also noted the difficulty of keeping rodents from entering and coming into contact with occupants. These findings are consistent with other studies that highlight the importance of household hygiene among members of hantavirus-endemic communities [22,42,43]. While hygiene is an effective and accessible preventive measure, it is limited in that it does not actively address vector control. Therefore, it is essential to establish mechanisms for consensus and dialogue that contribute to developing strategies explicitly focused on vector control.

Another relevant result of the study is the important role that the family plays in hantavirus health care and prevention. Family members adopt preventive practices against hantavirus, such as household hygiene, and encourage each other to seek timely access to health care when hantavirus is suspected. They also support and care for each other during illness and recovery. These findings align with those reported in the systematic review by Valencia Jiménez et al [44], which identified family involvement as an important factor in the prevention and control of zoonotic diseases such as dengue fever. The family is a key link in capacity building and community empowerment, grounded in respect for people's way of life, culture and social conditions. Other authors have also argued that the family fulfils important functions in health care, disease prevention, and the development of health-protective behaviours [45]. Health begins and is nurtured in the family. Public health professionals need to recognise this critical role of the family and include it in the planning, implementation and evaluation of health care and disease prevention programmes [46].

Our study also found that, in addition to the family, community members described two other important sources of support in coping with hantavirus: solidarity among neighbours and the wider community, and their belief in God and the power of prayer in providing spiritual strength. Several studies have shown that community cohesion and involvement in zoonotic disease prevention lead to greater acceptance of recommendations and improved outcomes [47,48]. For this reason, the cultural and social characteristics of communities must be central to the design of disease prevention initiatives [49]. Other studies focusing on other zoonotic diseases such as malaria [18] and dengue fever [44] have also demonstrated the importance of belief in and communication with a higher power or God for communities seeking protection and care in the face of disease. Similarly, a number of ethnonursing studies conducted in Panama have identified the importance of belief in God and the practice of prayer as resources for escaping cycles of domestic violence [50] and as mechanisms for coping with various everyday challenges [51].

The study findings emphasise the relevance of socio-cultural perspectives in understanding hantavirus disease in rural areas. We identified beliefs about the modes of transmission of the disease, its significance within family and community contexts, and associated fears as key dimensions in hantavirus prevention and control. Additionally, we found that religion and spirituality, along with the social and financial repercussions of the illness, play a significant role in how people confront and reinterpret the illness. Overall, these findings reveal the need for a holistic approach to hantavirus that incorporates social and cultural factors into prevention and control strategies. These findings align with those of other studies in different contexts, which also reveal the importance of socio-cultural perspectives in how hantavirus [9,52] and other zoonoses [17–21] are manifested, as well as with the One Health strategy, which adopts a similar approach [8,12,13].

## Implications for practice

Community members' insights into hantavirus disease (how they identify, describe and perceive it) are an essential source of knowledge for nurses and other health professionals, enabling them to develop hantavirus prevention programmes that cohere with the social and cultural perspectives of the community [15,24,29,30]. Integrating these perspectives into such programmes has been recognised as a priority by both anthropologists and advocates of the One Health strategy [10]. Within this context, ethnonursing and Leininger's Theory of Culture Care Diversity and Universality offer a deep understanding of the social and cultural dimensions of health. According to this theory, findings from ethnonursing-based studies are useful in developing decision and action modes for culturally congruent care in practice [29].

Based on the results of our study, we propose a model that adapts three culture care decision and action modes proposed by Leininger [53] to hantavirus prevention (Fig 2). Implementation of this model implies changes in behaviour, lifestyle and belief systems within the community. These changes do not occur spontaneously, but require effective communication, respect and consensus-building between the community, nurses and other health professionals. The proposed model is inherently sequential, starting with the mode that is most readily achievable and likely to generate maximum consensus, namely, the *Preservation and/or Maintenance of Culture Care* mode. This mode involves actions aimed at maintaining and promoting behaviours, lifestyles and values that are beneficial for hantavirus prevention. The second mode, termed *Accommodation/Negotiation of Culture Care*, presents greater challenges and necessitates joint efforts in communication and negotiation. This mode includes actions focused on negotiating and accommodating preventive practices that are either already partially implemented or that people are willing to adopt. Finally, the third mode in the model, *Repatterning/Restructuring of Culture Care,* is the most challenging to achieve, as it requires extensive negotiation and time. Actions within this mode aim to restructure or alter entrenched behaviours or lifestyles within the community that hinder hantavirus prevention.

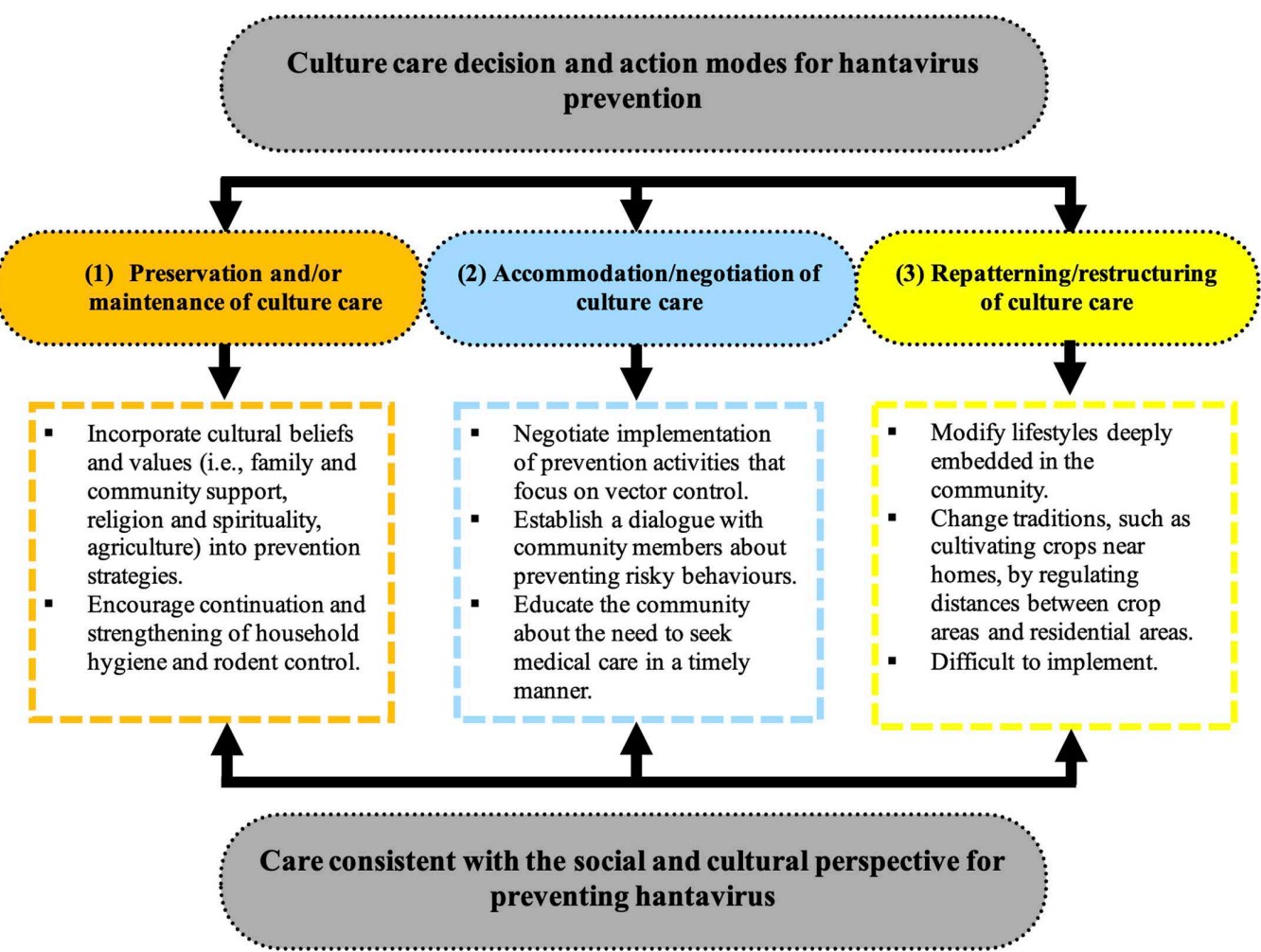

**Fig 2. Model of Culture Care Decision and Action Modes for Hantavirus Prevention, Adapted from Leininger.**

**Preservation and/or maintenance of culture care.** A first set of actions to be implemented by nurses and other health professionals relates to respecting the beliefs and values of the community and finding ways to integrate them into the prevention strategies being implemented. First, health professionals could identify the cultural elements that underpin family life in the community, namely the ways in which they express interest in, support and pursue the welfare and care of others, both within the family and in the wider community. These values of support and solidarity could be used by professionals to promote collaboration among community members in defining and implementing hantavirus prevention practices. Second, nurses and other health professionals could seek to understand the significance of religion and spirituality in the lives of community members in order to enlist the support of religious and spiritual leaders in providing health education for hantavirus prevention. Third, when developing prevention strategies, professionals could consider the importance of agriculture and rural work to community members, a value that has been passed down through generations of families. Greater cultural understanding and sensitivity on the part of nurses and other health professionals with regard to the value of agricultural life to the community would help foster closer and more collaborative relationships, ultimately improving the effectiveness of proposed prevention strategies. Fourth, nurses and other health professionals could encourage the continuation and strengthening of evidence-based preventive practices mentioned by participants, such as household hygiene and rodent control, including the use of domestic cats to hunt the vector.

**Accommodation/Negotiation of Culture Care.** This second set of actions for nurses and other health professionals emphasises the negotiation of hantavirus prevention measures with community members. First, given that two of the three beliefs expressed by participants about the causes of the disease were directly or indirectly related to the vector, health professionals could seek to negotiate a focus on vector control in prevention activities. These measures may include sealing rodent entry points into the home, reducing access to food sources, and setting traps or using poison to catch and eliminate mice. Second, health professionals could create the conditions for open and respectful dialogue with community members about other behaviours that put them at risk of hantavirus transmission. These behaviours include improper storage of harvested food inside the home or in open containers that allow rodents access and the limited use of face masks and gloves when handling live or dead rodents, or when entering or cleaning enclosed spaces, storage areas or unoccupied houses. Any dialogue aimed at changing these behaviours must consider the availability and management of resources such as gloves, masks and food storage containers. In addition, open meetings, community discussions and educational activities could be organised to promote timely seeking of medical care in response to the risk of hantavirus and to emphasise its benefits in terms of patient survival and recovery.

**Repatterning/restructuring of culture care.** This third set of actions aims to address practices and lifestyles that have been handed down through generations and are deeply engrained in the community, where they pose a risk of hantavirus transmission. These practices are harder to change because they are deeply embedded in the cultural group. One example is crop cultivation in close proximity to homes, which increases the likelihood of contact with the rodent vector of hantavirus. Despite the need for authorities to regulate appropriate distances between crop fields and residential areas, such initiatives are difficult to implement because of resistance rooted in culture and tradition.

## Limitations and future research

This study has several limitations. First, the results are limited to the community of Bebedero in Tonosí, located in the province of Los Santos, Panama, and therefore cannot be generalised to other endemic communities. However, it is important to note that this community has experienced high hantavirus incidence and mortality over several decades, suggesting that it may serve as a representative example of an endemic community. In addition, the procedures and results were reported in extensive detail to promote "thick description" [54], thereby enabling others to assess the transferability of the study's conclusions to different contexts. Second, as data collection took place during the COVID-19 pandemic, the immersion process was interrupted and resumed in alignment with public health regulations. This circumstance may have contributed to the delayed acceptance of the researcher by some participants. Third, involving key participants from

the quantitative phase in the qualitative phase may have introduced bias into their responses, given that they had already been exposed to this topic. However, interviewing participants who have already taken part in a previous quantitative phase is a legitimate strategy in explanatory sequential mixed methods designs, as it allows the quantitative findings to be explored in more depth. In addition, the selected participants had close personal and familial experience with hantavirus. Therefore, regardless of their prior participation in the survey, these individuals had solid knowledge of and experience with the topic, making it highly likely that their responses would have been similar had they not completed the survey. Fourth, due to public health restrictions, field observations focused primarily on interactions within the family context, with fewer opportunities to observe interactions at the community level. Fifth, also as a result of the public health restrictions, some interviews with key participants were shorter than originally planned (in some cases 30 minutes instead of the planned 60 minutes), necessitating a second interview. Furthermore, the necessity to conduct interviews with general participants in a virtual format resulted in reduced interaction and constrained the ability to comprehensively capture physical expressions and non-verbal communication. One final relevant limitation was the low participation of men among the key participants, which may have constrained the depth of the analysis from a gender perspective.

The results of this study may serve as a basis for future research. The study could be replicated in other hantavirus-endemic areas of Panama, allowing for comparative analysis of results. In addition, future research could evaluate the effectiveness and acceptability of implementing a prevention programme that integrates the model of culture care decision and action modes for hantavirus prevention described above.

## Conclusion

This study contributes to the existing body of knowledge regarding the importance of beliefs and socio-cultural perspectives in the care and prevention of diseases such as zoonoses. The study provides an in-depth exploration of the community's beliefs about the modes of hantavirus transmission and the care and prevention practices adopted in accordance with these beliefs. Additionally, it documents the community's fears, concerns and suggestions for addressing the hantavirus disease. The findings highlight the importance and significance of agricultural work in the rural community under study, as well as the significance of religious and social community celebrations. They also underscore the value of family and community as sources of support in daily life and health care in the face of the risk of hantavirus infection. In the spiritual realm, belief in God as a higher power offering protection from disease and strength in times of illness was identified, along with the value of individual and communal prayer as a means of seeking divine protection from disease. Overall, this study provides valuable insights for the development of hantavirus prevention education programmes that align with the social and cultural perspectives of community members. We suggest adopting the model of culture care decision and action modes proposed in this study to enhance active community involvement in designing and implementing hantavirus prevention programmes. From a sociocultural perspective, this approach ensures that prevention programmes are relevant, sustainable and culturally appropriate for the affected communities.

## Supporting information

**S1 File. Interview Guide.**
(DOCX)

**S2 File. Consolidated Criteria for Reporting Qualitative Research (COREQ) Checklist.**
(PDF)

**S3 File. Inclusivity in Global Research Questionnaire.**
(DOCX)

**S4 File. Key and General Participants Characteristics.**

(DOCX)

## Acknowledgments

The authors would like to express their gratitude to the participants for their valuable contributions, without which this study would not have been possible. They would also wish to thank the community leader, Leysi Yovana Villarreal, for the support she provided to the principal investigator in facilitating immersion in the community and communication with participants, which were crucial to the success of this study. The authors would like to acknowledge the work of a professional translator and editor in translating and editing some parts of the original draft of the manuscript. They would also like to acknowledge the use of the Paperpal tool for grammatical and linguistic improvements to the final version of this manuscript.

## Author contributions

**Conceptualization:** Janeth Agrazal Garcia, Lydia Gordón de Isaacs.

**Data curation:** Janeth Agrazal Garcia.

**Formal analysis:** Janeth Agrazal Garcia.

**Funding acquisition:** Janeth Agrazal Garcia.

**Investigation:** Janeth Agrazal Garcia.

**Methodology:** Janeth Agrazal Garcia, Lydia Gordón de Isaacs, Elsa Lucia Escalante-Barrios, Sergi Fàbregues.

**Project administration:** Janeth Agrazal Garcia.

**Supervision:** Janeth Agrazal Garcia, Elsa Lucia Escalante-Barrios, Sergi Fàbregues.

**Writing – original draft:** Janeth Agrazal Garcia, Lydia Gordón de Isaacs, Elsa Lucia Escalante-Barrios, Sergi Fàbregues.

**Writing – review & editing:** Janeth Agrazal Garcia, Lydia Gordón de Isaacs, Elsa Lucia Escalante-Barrios, Sergi Fàbregues.

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
