## [Decision Letter · Decision Letter 0]

17 Jul 2025

PGPH-D-25-01269

Beliefs and socio-cultural perspectives on hantavirus in a rural community in Panama: an ethnonursing study

Dear Dr. Agrazal,

Thank you for submitting your manuscript to PLOS Global Public Health. After careful consideration, we feel that it has merit but does not fully meet PLOS Global Public Health’s publication criteria as it currently stands. Therefore, we invite you to submit a revised version of the manuscript that addresses the points raised during the review process.

We look forward to receiving your revised manuscript.

Kind regards,

Najmul Haider, PhD

Academic Editor

Journal Requirements:

Additional Editor Comments (if provided):

Reviewers' comments:

Reviewer's Responses to Questions

**Comments to the Author**

1. Does this manuscript meet PLOS Global Public Health’s publication criteria ? Is the manuscript technically sound, and do the data support the conclusions? The manuscript must describe methodologically and ethically rigorous research with conclusions that are appropriately drawn based on the data presented.

Reviewer #1: Yes

Reviewer #2: Yes

2. Has the statistical analysis been performed appropriately and rigorously?

Reviewer #1: N/A

Reviewer #2: N/A

3. Have the authors made all data underlying the findings in their manuscript fully available (please refer to the Data Availability Statement at the start of the manuscript PDF file)?

Reviewer #1: No

Reviewer #2: No

4. Is the manuscript presented in an intelligible fashion and written in standard English?

Reviewer #1: Yes

Reviewer #2: Yes

5. Review Comments to the Author

Reviewer #1: Abstract

The host of hantavirus could be briefly mentioned to better align the abstract with the socio-ecological drivers that facilitate spillover.

The study timeline is missing and should be included.

More detail is needed on participant selection: Who were the participants, how were they selected, and why? It is not immediately clear who were considered general participants versus key participants.

The selection criteria for key participants should be explained—participation in the prior quantitative study alone does not sufficiently justify their inclusion.

A brief recommendation should be added to the abstract for completeness.

Main Manuscript

Introduction

There is inconsistency in terminology: the abstract refers to "Hantavirus Pulmonary Syndrome" while the introduction uses "Hantavirus Cardiopulmonary Syndrome". Please ensure consistent terminology throughout.

While the authors mention the socio-cultural dimensions of hantavirus, additional justification is needed in the introduction to explain how this study adds to the existing literature.

Leininger’s Theory of Culture Care Diversity and Universality is mentioned multiple times and could be referenced more concisely.

The ethnonursing approach should be elaborated further to clarify its application.

Methods

The study timeline (September 3, 2021, to August 25, 2022) should be moved to the beginning of the Methods section.

The use of Leininger’s Stranger to Trusted Friend Enabler needs more elaboration—how and why was it applied?

Provide more clarity on key participant selection criteria, including why they were selected and how their involvement contributes uniquely to the study.

A good description is provided for general participants.

The process of theoretical saturation should be explained. Was data analysed during fieldwork, and what strategies were used to determine when saturation was reached?

Results

Under participant characteristics, it may not be necessary to separate key participants and general participants in the analysis—they could be presented together unless a clear justification is provided for treating them separately.

Discussion

The discussion is well-structured; however, in several places, the results are repeated without deeper interpretation. Consider focusing on interpreting the findings rather than reiterating them.

The authors should explain why participants hold certain beliefs and discuss their implications for risk perception, prevention, and control.

The study site name, currently mentioned in the discussion, should be included earlier in the Methods section.

Please address the potential bias introduced by selecting participants from the prior quantitative study. Their prior exposure to structured questions and potential discussions with peers may have influenced their knowledge or perception of hantavirus.

Recommendations should be more specific, particularly regarding what is needed to address gaps in the existing health infrastructure.

Overall Comment:

This is an excellent and important study. With some revisions for clarity, consistency, and elaboration in key areas, the manuscript will make a valuable contribution to the literature on hantavirus and community-based health perspectives.

Reviewer #2: The article provides a very well-written depiction of community’s beliefs about the modes of hantavirus transmission and the care and prevention practices adopted in accordance with these beliefs. Most importantly, the authors leveraged the findings to provide important insights for the development of hantavirus prevention education programmes aligning with the social and cultural beliefs and norms, and the socio-economic context of community members.

I only have some minor comments for further clarification.

1. I am not confident about how well-balanced the amount of details in theme 2, particularly pattern 2.2, is in respect to its interpretation in the discussion and direct link with the understanding of issues related to hantavirus disease. The fifth paragraph of the discussion section might be revisited to more explicitly interpret the linkage between hantavirus infection and community’s practice and perception related to agricultural and rural work in light of findings in other contexts.

2. Gender perspective seems to be missing. Given the gender imbalance in key participants (mostly women), I am afraid that there would be important perspective missed from men, who are often important role players or decision makers in many rural communities. It should at least be addressed in the limitations.

3. Methods section could include information on how participants were recruited based on age or gender.

4. The discussion and conclusion sections could be further improved by minimizing repetition of results and incorporating more critical discussion in light of situation in other countries/context.

5. Instead of including description of individual participants, table 1 and 2 could be combined to present summarized description.

6. PLOS authors have the option to publish the peer review history of their article (what does this mean? ). If published, this will include your full peer review and any attached files.

**Do you want your identity to be public for this peer review?** For information about this choice, including consent withdrawal, please see our Privacy Policy .

Reviewer #1: **Yes: ** Md Saiful Islam

Reviewer #2: No

---

## [Decision Letter · Decision Letter 1]

26 Sep 2025

Beliefs and socio-cultural perspectives on hantavirus in a rural community in Panama: an ethnonursing study

PGPH-D-25-01269R1

Dear Dr. Agrazal,

We are pleased to inform you that your manuscript 'Beliefs and socio-cultural perspectives on hantavirus in a rural community in Panama: an ethnonursing study' has been provisionally accepted for publication in PLOS Global Public Health.

Best regards,

Najmul Haider, PhD

Academic Editor

Reviewer Comments (if any, and for reference):

Reviewer's Responses to Questions

**Comments to the Author**

1. If the authors have adequately addressed your comments raised in a previous round of review and you feel that this manuscript is now acceptable for publication, you may indicate that here to bypass the “Comments to the Author” section, enter your conflict of interest statement in the “Confidential to Editor” section, and submit your "Accept" recommendation.

Reviewer #2: All comments have been addressed

2. Does this manuscript meet PLOS Global Public Health’s publication criteria ? Is the manuscript technically sound, and do the data support the conclusions? The manuscript must describe methodologically and ethically rigorous research with conclusions that are appropriately drawn based on the data presented.

Reviewer #2: Yes

3. Has the statistical analysis been performed appropriately and rigorously?

Reviewer #2: N/A

4. Have the authors made all data underlying the findings in their manuscript fully available (please refer to the Data Availability Statement at the start of the manuscript PDF file)?

Reviewer #2: No

5. Is the manuscript presented in an intelligible fashion and written in standard English?

Reviewer #2: Yes

6. Review Comments to the Author

Reviewer #2: The comments provided have been addressed adequately. I only have a few minor suggestions for consideration.

Abstract: The following sentence in abstract needs to be revised as follows by adding a few words that are missing perhaps.

"The disease is transmitted to humans through the inhalation of aerosols containing 'viruses from' feces, urine, and saliva from asymptomatic infected rodents.”

Methods: The timeline is repeated later in the data collection section.

7. PLOS authors have the option to publish the peer review history of their article (what does this mean? ). If published, this will include your full peer review and any attached files.

**Do you want your identity to be public for this peer review?** For information about this choice, including consent withdrawal, please see our Privacy Policy .

Reviewer #2: No
